# Loperamide Inhibits Replication of Severe Fever with Thrombocytopenia Syndrome Virus

**DOI:** 10.3390/v13050869

**Published:** 2021-05-10

**Authors:** Shuzo Urata, Jiro Yasuda, Masaharu Iwasaki

**Affiliations:** 1National Research Center for the Control and Prevention of Infectious Diseases (CCPID), Nagasaki University, 1-12-4 Sakamoto, Nagasaki 852-8523, Japan; j-yasuda@nagasaki-u.ac.jp; 2Department of Emerging Infectious Diseases, Institute of Tropical Medicine (NEKKEN), Nagasaki University, 1-12-4 Sakamoto, Nagasaki 852-8523, Japan; 3Laboratory of Emerging Viral Diseases, International Research Center for Infectious Diseases, Research Institute for Microbial Diseases, Osaka University, 3-1 Yamadaoka, Osaka 565-0871, Japan; miwasaki@biken.osaka-u.ac.jp

**Keywords:** severe fever with thrombocytopenia syndrome virus, antiviral, loperamide

## Abstract

Background: Severe fever with thrombocytopenia syndrome (SFTS) is an emerging tick-borne infectious disease caused by the SFTS virus (SFTSV). SFTS is mainly prevalent in East Asia. It has a mortality rate of up to 30%, and there is no approved treatment against the disease. In this study, we evaluated the effect of loperamide, an antidiarrheal and antihyperalgesic agent, on the propagation of SFTSV in a cell culture system. Methods: SFTSV-infected human cell lines were exposed to loperamide, and viral titers were evaluated. To clarify the mode of action of loperamide, several chemical compounds having shared targets with loperamide were used. Calcium imaging was also performed to understand whether loperamide treatment affected calcium influx. Results: Loperamide inhibited SFTSV propagation in several cell lines. It inhibited SFTSV in the post-entry step and restricted calcium influx into the cell. Furthermore, nifedipine, a calcium channel inhibitor, also blocked post-entry step of SFTSV infection. Conclusions: Loperamide inhibits SFTSV propagation mainly by restraining calcium influx into the cytoplasm. This indicates that loperamide, a Food and Drug Administration (FDA)-approved drug, has the potential for being used as a treatment option against SFTS.

## 1. Introduction

Severe fever with thrombocytopenia syndrome (SFTS) was found in China in 2011 and has been reported thereafter in several East Asian countries, including Japan, South Korea, Taiwan, and Vietnam [1,2,3,4,5,6]. SFTS is an emerging infectious disease caused by the SFTS virus (SFTSV, SFTS phlebovirus, or Huaiyangshan banyangvirus). SFTSV is a tick-borne virus and is classified into the genus *Banyangvirus* of the family *Phenuiviridae*, order *Bunyavirales* [7]. Other phleboviruses that are phylogenetically related to SFTSV, namely Heartland virus and Malsoor virus, were isolated from Missouri, USA, and western India, respectively [8,9]. In addition, a novel virus closely related to SFTSV and Heartland virus was identified in China [10]. At present, there is no established prophylaxis or treatment against SFTS. It is critical to develop such prophylaxis/treatment, and elucidation of the virus replication mechanism could help determine treatment strategies. Despite the lack of established treatments against SFTS, several compounds, including Food and Drug Administration (FDA)-approved drugs, have been reported to inhibit SFTSV replication both in vivo and in vitro [11]. In this study, we focused on the FDA-approved antidiarrheal drug loperamide because diarrhea is a prominent symptom of SFTS [5]. Intriguingly, loperamide has shown to inhibit multiplications of several human coronaviruses including severe acute respiratory syndrome coronavirus 2 (SARS-CoV-2), which also cause diarrhea in humans [12,13]. In addition, loperamide was proposed to exert its antidiarrheal effect by blocking calcium channels [14] and several calcium channel blockers have been reported to inhibit the SFTSV replication in vivo and in vitro [11]. These findings encourage efforts to investigate the potential clinical use of loperamide to reduce viral load and simultaneously to alleviate diarrhea in SFTS patients. In this study, we demonstrated that loperamide inhibited SFTSV propagation in two human cell lines and in a simian cell line, and its main target is the post-entry step due to the inhibition of the calcium influx.

## 2. Materials and Methods 

### 2.1. Cells, Viruses, and Materials

Huh-7, Vero 76, and SW13 cells were maintained in Dulbecco’s Modified Eagle’s Medium (DMEM) supplemented with 1% penicillin-streptomycin and 10% fetal bovine serum (FBS), as described previously [15]. The source of SFTSV (YG-1) and production of polyclonal antibodies against SFTSV N protein have been described previously [15]. Rhod-4 AM (21121), used as the calcium indicator, was obtained from AAT Bioquest (Sunnyvale, CA, USA). Loperamide (L0154) was purchased from Tokyo Chemical Industry (Tokyo, Japan). Amantadine (21364), ivabradine (15868), naloxone (15594), and nifedipine (11106) were obtained from Cayman (Ann Arbor, MI, USA).

### 2.2. Virus Infection and Treatment with Compounds

Confluent cell monolayers of Huh-7, SW13, or Vero 76 cells (2 × 10^4^ cells) were infected with SFTSV, at a multiplicity of infection (MOI) of 0.1, in 96-well plates. Fresh medium containing different concentrations of the compounds was added to wells containing the infected cells. At 24 and 48 h post infection (h p.i.), the culture supernatants were collected, and virus titers were measured as per the procedure described below (Huh-7 and SW13 cells). In case of Vero76 cells, infected cells were fixed and stained for SFTSV N at 24 h p.i.

### 2.3. Viral Titration

The SFTSV titer was determined using an immunofocus assay. Vero 76 cells (3 × 10^4^ cells/well) were seeded in a 96-well plate 1 day prior to infection. The cells were infected with 1:10 serial dilutions of the virus and incubated for 16 h at 37 °C in 5% CO_2_. The cells were fixed with 4% paraformaldehyde (PFA) for 30 min at 15–25 °C, and then incubated in PBS containing 0.1% Tween^®^20 for 1 h at 15–25 °C. The cells were blocked with a solution of 10% FBS in dilution buffer (3% BSA, 0.3% Triton X-100/PBS (−)) at 4 °C overnight. SFTSV N protein was detected using a primary anti-SFTSV N antibody, followed by a secondary anti-rabbit IgG-FITC antibody (ab6009, Abcam, Cambridge, UK). SFTSV N-positive cells were counted and calculated as fluorescent focus units (FFU/mL).

### 2.4. Time-of-Addition Infection Assay

Huh-7 cells (2 × 10^4^/well) were seeded in a 96-well plate. Culture medium in the wells was replaced with fresh medium containing loperamide (20 μM), and the cells were incubated for 1 h (pre-entry treatment (Pre-)). For the remaining samples (Ctrl, During-, and Post-), the culture medium was replaced with fresh medium. All samples were infected with SFTSV at an MOI of 1 for 1 h. For the during-entry treatment (During-), loperamide (final concentration 20 μM) was included in media containing the virus. After the viral infection step, the culture medium was replaced with fresh medium containing either DMSO (Ctrl, Pre-, and During-) or 20 μM loperamide (post-entry treatment (Post-)). After incubation for 16 h, the cells were fixed with 4% PFA and stained with an anti-SFTSV N antibody. Images were captured using a BZ-X700 microscope (Keyence, Osaka, Japan). 

### 2.5. Counting Fluorescent (SFTSV N-Positive) and DAPI-Stained Cell Numbers

SFTSV N-positive and DAPI-stained cells, captured by BZ-X700, were counted automatically using a BZ-X Analyzer (Keyence), and from four independent fields belonging to more than three independent wells. For SFTSV N-positive and DAPI-positive cell numbers, relative positive cell numbers were calculated, normalized with the control (DMSO) cell number, and presented as the mean ± standard deviation (SD). 

### 2.6. Calcium Imaging and Analysis

Huh-7 cells (2 × 10^4^/well) were seeded in 96-well plates and treated with either DMSO or 10 μM loperamide in the presence or absence of SFTSV (MOI = 0.1). After 48 h of treatment, the culture medium was replaced with Rhod-4AM (final concentration of 10 μM) diluted in Opti-MEM and incubated for 20 min at 37 °C in 5% CO_2_. After incubation, the cells were fixed with 4% PFA. Fixed cells were imaged (fluorescent and bright fields) using BZ-X700. Fluorescence intensity from three independent wells was measured using SpectraMAX iD5 (Molecular Device, San Jose, CA, USA).

### 2.7. Statistical Analysis

Excel and GraphPad Prism 5 (GraphPad Software, Inc., San Diego, CA, USA) software were used for all statistical analyses. Quantitative data were presented as the mean ± SD from at least three independent experiments (unless indicated otherwise). For all calculations, *p* < 0.05, was considered significant and was represented using an asterisk (*). Group comparisons were performed using one-way analysis of variance (ANOVA), followed by Dunnett’s multiple comparison test. Welch’s *t*-test was used to compare the two groups.

## 3. Results

### 3.1. Loperamide Treatment Inhibited SFTSV Propagation

It was reported that SFTSV-infected cells could be detected in several organs from the patients, including liver and adrenal gland [16]. To examine if loperamide has an anti-SFTSV effect *in vitro*, Huh-7 (human liver originated cell line) and SW13 (human adrenal cortex originated cell line) cells, in which apparent cytopathic effect upon SFTSV infection was not observed [15], were infected with SFTSV (MOI = 0.1), followed by loperamide or DMSO (control) treatment. At 24 and 48 h p.i., culture supernatants were collected to measure virus titers. In case of both cell lines, virus titers were significantly reduced in supernatants of loperamide-treated samples in comparison to those for the DMSO-treated controls (Figure 1A,B). In Huh-7 cells, loperamide treatment reduced SFTSV titers approximately 3-fold and 24-fold at 24 and 48 h p.i., respectively (Figure 1A). In SW13 cells, when compared to the controls, loperamide treatment reduced SFTSV titers approximately 3-fold and 90-fold at 24 and 48 h p.i., respectively (Figure 1B). To determine if the reduction in SFTSV titers was due to the cytotoxic effects of loperamide, Huh-7 cells infected with SFTSV shown in Figure 1A were fixed and stained with DAPI for both 24 h (Figure 1C) and 48 h (Figure 1D) post infection. The relative percentages of DAPI-positive Huh-7 cells 24 and 48 h post loperamide treatment were 89.1% and 88.6%, respectively. The 50% inhibitory concentration (IC_50_) of loperamide against SFTSV was also evaluated in Huh-7 cells. Huh-7 cells were infected with SFTSV (MOI = 0.1) for 1 h and loperamide was administered at concentrations of 1, 5, 10, and 20 μM for 48 h. Cells treated with DMSO instead of loperamide were used as controls. Culture supernatants were collected to measure virus titers and the IC_50_ of loperamide was found to be 4.4 μM (Figure 1E).

### 3.2. Loperamide Inhibited Post-Entry Step, but Not Pre- and during-Entry Stages, of SFTSV Infection

To determine the inhibitory step of loperamide on SFTSV infection, a time-of-addition infection assay was performed in Huh-7 cells (Figure 2A). An image representative of SFTSV N-positive cells across the different stages is shown in Figure 2B, while Figure 2C shows the average number of SFTSV N-positive cells per field. Pretreatment with loperamide did not affect N-positive cell number, and during-treatment of loperamide led to modest increase in the number of N positive cells, when compared with the DMSO control. In contrast, post-treatment with loperamide significantly reduced (four- to five-fold reduction) the number of SFTSV N-positive cells compared to the control treatment. This result indicated that loperamide targets the post-entry, which includes the replication, transcription, and translation, but not the pre- and during-entry steps, of SFTSV infection.

### 3.3. Nifedipine, but Not Ivabradine, Amantadine, or Naloxone, Inhibited Post-Entry Step of SFTSV Infection

Loperamide targets L- and T-type calcium channels [14,17], hyperpolarization-activated cyclic nucleotide-gated (HCN) channels [18,19,20], N-methyl-D-aspartate (NMDA) receptors [21], and μ-opioid receptors [22,23,24]. To determine the target of loperamide for inhibiting the post-entry step of SFTSV infection, several small chemical compounds known to specifically inhibit the shared aforementioned targets with loperamide were used. Nifedipine inhibits L-type calcium channels. Ivabradine is known to target HCN channels [19]. Amantadine is known to inhibit NMDA receptors [25]. Naloxone is a μ-opioid receptor antagonist [26,27]. First, we examined the cytotoxic effects of these compounds by counting the number of DAPI-positive cells (Figure 3A) after treatment with the above-mentioned compounds. None of the compounds exhibited significant cytotoxic effects in Huh-7 cells at the concentrations used in this study. Huh-7 cells were infected with SFTSV (MOI = 0.1) and the compounds listed above were administered (nifedipine (250 μM), ivabradine (250 μM), amantadine (625 μM), or naloxone (1 mM)). At 16 h p.i., infected cells were fixed. Fixed cells were then stained with anti-SFTSV N-antibody, and the number of N-positive cells was counted and analyzed (Figure 3B). DMSO-treated control cells and ivabradine, amantadine, and naloxone-treated cells had equivalent numbers of SFTSV N-positive cells. In contrast, the number of SFTSV N-positive cells in nifedipine-treated wells was reduced significantly; it approximately halved in comparison to that for the DMSO control. Naloxone is known to antagonize loperamide; therefore, we evaluated the effect that co-administering naloxone and loperamide (Figure 3C) would have on the SFTSV N-positive cell numbers. Loperamide alone and in combination with naloxone was found to reduce the SFTSV N-positive cell number equivalently. To examine if the anti-SFTSV effect of loperamide was related to the anti-viral response including the type I interferon signaling, Vero 76 cells, which is known to be deficient for the type I interferon production, was infected with SFTSV and treated with and without loperamide. As shown in Figure 3D, the relative N positive cell number upon loperamide treatment was significantly reduced (five-fold) compared to that of the DMSO treatment.

### 3.4. Calcium Influx Was Inhibited by Loperamide Treatment in Huh-7 Cells

Our previous experiment showed that nifedipine inhibited the post-entry step of SFTSV infection (Figure 3B). To examine if loperamide indeed affects calcium influx in Huh-7 cells, calcium imaging and analysis were performed. Representative images were presented in Figure 4A and the analysis in Figure 4B. The calcium probe (Rhod-4AM), in red, was located in the cell cytoplasm. Relative fluorescence intensity was measured, and we found that loperamide treatment significantly reduced the fluorescence intensity to 40%. Calcium influx was also examined in the presence of the SFTSV infection (Figure 4C,D). Loperamide treatment reduced the calcium influx approximately five-fold compared to DMSO treatment (Figure 4C). The culture supernatant from the samples used in Figure 4C was used to measure the virus production. Loperamide treatment reduced SFTSV titers approximately six-fold (Figure 4D).

## 4. Discussion

Due to its high fatality rate and lack of approved prophylaxis and drugs, SFTS was listed by the World Health Organization among the top 10 priority infectious diseases with an urgent need for established treatment [28]. Since its discovery in China in 2011, many efforts have been made to identify effective drugs to treat SFTS [11]. We have previously shown that several chemical compounds that restrain fatty acid and cholesterol synthesis (fenofibrate and lovastatin) inhibited SFTSV propagation [15]. One of the prominent small chemical compounds is T-705 (favipiravir), whose anti-SFTSV effect was observed both in vitro [29] and in vivo [30]. The anti-SFTSV effects of the chemical compounds ribavirin [31,32], caffeic acid [33], amodiaquine [34], hexachlorophene [35], and 2′-FdC [36] and those of the biologics interferon-α, -β, and -γ [32,37] have been demonstrated previously. Recently, the anti-SFTSV effects of several catechins and flavonols from green tea [38] and NF-κB inhibitor SC75741 [39] have also been reported. Calcium channel inhibitors, drugs against high blood pressure such as benidipine and nifedipine, were also reported to reduce SFTSV propagation both in vitro and in vivo [40]. To identify additional drugs or compounds that might inhibit SFTSV propagation, we focused on loperamide, an FDA-approved anti-diarrhea drug. In the current study, we showed that loperamide reduced SFTSV propagation in two human cell lines (Huh-7 and SW13) and had an IC_50_ of 4.4 μM in Huh-7 cell line (Figure 1). Although the treatment of loperamide in Huh-7 cells for 48 h at 20 μM affected to the cell viability, slightly but statistically significantly, 25-fold reduction of the SFTSV production was observed from the loperamide-treated cells compared to the DMSO-treated cells, concluding to us that loperamide inhibited SFTSV propagation in Huh-7 cells. Previous study reported that the IC_50_ of loperamide against Middle East respiratory syndrome coronavirus (MERS-CoV) is 3–8 μM [12]. Similar IC_50_ values between SFTSV and MERS-CoV implied parallels in the anti-viral effect of loperamide. To ascertain how loperamide inhibited post-entry step of SFTSV infection, a time-of-addition infection assay was performed (Figure 2). Although the treatment of loperamide during the infection increased the relative SFTSV N positive cell numbers slightly, but statistically significantly (1.22 times compared to Ctrl.), we focused on the significant reduction of the SFTSV N positive cell number upon the post-treatment (3.6-fold reduction compared to Ctrl.). These experiments indicated that the main target of loperamide is the post-entry step, rather than the pre- or during-entry step.

Although small chemical compounds are designed to specifically bind to a target, many compounds could affect non-primary targets. Loperamide has been reported to affect L-type and T-type calcium channels, HCN channels, NMDA receptors, and μ-opioid receptors. To identify the main target of loperamide for inhibiting post-entry step of SFTSV infection, small chemical compounds targeting the L-type calcium channels (nifedipine), HCN channel (ivabradine), NMDA receptor (amantadine), μ-opioid receptor (naloxone) were used (Figure 3). Nifedipine inhibited post-entry step of SFTSV infection (Figure 3B) similar to loperamide (Figure 2), suggesting that calcium influx was involved in the post-entry step of SFTSV infection, consistent with a previous report [40]. Since the IC_50_ of ivabradine for HCN channels was approximately 2 μM in HEK-293 cells [19], and ivabradine treatment in our study (250 μM) did not affect the post-entry step of the SFTSV infection, we concluded that HCN channels are not involved in post-entry step of SFTSV infection (Figure 3B). Amantadine is known to antagonize the NMDA receptor and is used to treat Parkinson’s disease. The antiviral effect of amantadine has also been reported for several viruses. For example, the anti-influenza A virus activity of amantadine was reported to be 1–10 μM by inhibiting M2 ion channel activity [41,42,43]. Similarly, ion channel activities of Chikungunya virus 6 K [44], hepatitis C virus p7, human immunodeficiency virus type 1 Vpu, and picornavirus 2 B, were also reported to be inhibited by amantadine [45]. The anti-dengue virus effect of amantadine was also reported at a concentration of 50 μg/mL (=330 μM) (IC_90_), which is close to 250 mM that we used in the study [46]. Naloxone was reported to affect neural stem cells via a receptor-independent pathway [47,48], suggesting that naloxone alone could affect the cells. Naloxone was also reported to antagonize the activation of loperamide-mediated μ-opioid receptor signaling [26]. However, in our study, naloxone alone did not affect the post-entry step of SFTSV infection. Additionally, our study also revealed that naloxone does not antagonize the loperamide-mediated reduction of the post-entry step of SFTSV infection, thereby suggesting that μ-opioid receptors are not involved in the post-entry step of SFTSV infection in Huh-7 cells. 

To explore if the anti-SFTSV effect of loperamide is the result of the host anti-viral response or not, Vero 76 cells, which is known to be deficient for the type I interferon (IFN) production, were infected with SFTSV and treated with and without loperamide. The significant reduction of the N positive cells upon loperamide treatment compared to the control treatment, strongly suggested that the main anti-SFTSV effect of loperamide was not due to the type-I IFN response.

Since nifedipine inhibited the post-entry step of SFTSV infection in Huh-7 cells, it was speculated that loperamide also inhibited the post-entry step of SFTSV infection by inhibiting calcium influx in Huh-7 cells. To assess this, calcium imaging was used to monitor calcium influx upon loperamide treatment in Huh-7 cells (Figure 4). In Huh-7 cells treated with DMSO control, calcium-dependent fluorescence was detected in almost all cells. In contrast, when the cells were treated with loperamide, florescence was detected only in a few cells. Quantification of the fluorescence intensity revealed that compared to DMSO, loperamide treatment significantly reduced the fluorescence intensity. Similar results were observed in the presence of the SFTSV (Figure 4C). These results indicated that loperamide inhibited calcium influx in Huh-7 cells.

Taken together, loperamide blocks post-entry step of SFTSV infection and propagation, and one of the mechanisms underlying its anti-SFTSV effects was the inhibition of calcium influx. In the past, a few studies have demonstrated the effect of loperamide on virus propagation or virus-related symptoms. As an antiviral agent, loperamide was found to inhibit MERS-CoV [12] and SARS-CoV-2 [13] replications through FDA-approved compound library screening, although its mechanism of action was not examined. Loperamide was also described to improve herpes simplex virus type-1 induced allodynia through the stimulation of μ-opioid receptors [49]. Furthermore, a perspective review proposed the use of loperamide for the treatment of voluminous diarrhea caused by the Ebola virus disease [50]. These observations suggested loperamide as a broad spectrum antiviral and symptom-improving agent. 

## Figures and Tables

**Figure 1 viruses-13-00869-f001:**
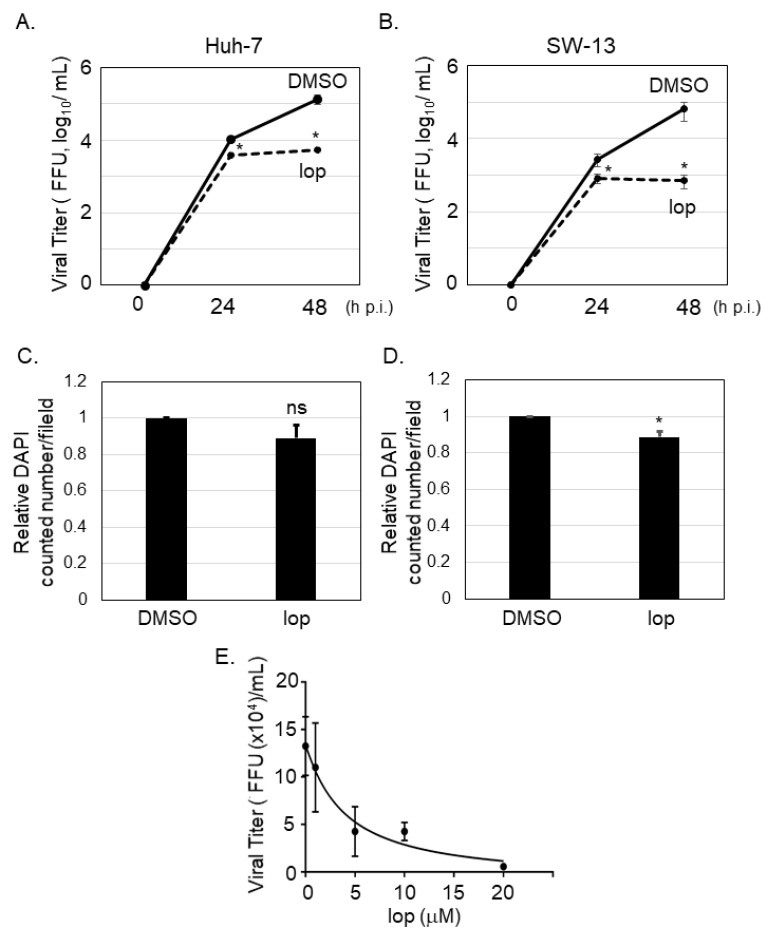
Loperamide treatment inhibited severe fever with thrombocytopenia syndrome virus (SFTSV) propagation. Huh-7 (**A**) or SW-13 (**B**) cells were infected with SFTSV (multiplicity of infection (MOI) = 0.1) and treated with loperamide (lop, 20 μM). Culture supernatant was collected at 24 and 48 h post infection (h p.i.) to measure the virus titers using Vero 76 cells. (**C**,**D**) Infected cells were fixed with 4% paraformaldehyde (PFA) and stained with DAPI for counting the numbers of viable cells. Relative DAPI-positive cell numbers from Huh-7 cells at 24 h (**C**) and 48 h (**D**) post loperamide treatment, respectively. (**E**) 50% inhibitory concentration (IC_50_) of loperamide against SFTSV in Huh-7 cells was calculated. Huh-7 cells were infected with SFTSV at an MOI = 0.1 and treated with 1, 5, 10, or 20 μM of loperamide. At 48 h p.i., culture media was collected to measure SFTSV titer. IC_50_ was calculated using GraphPad Prism 5. Data correspond to the mean ± SD (ns; not significant, * *p* < 0.05).

**Figure 2 viruses-13-00869-f002:**
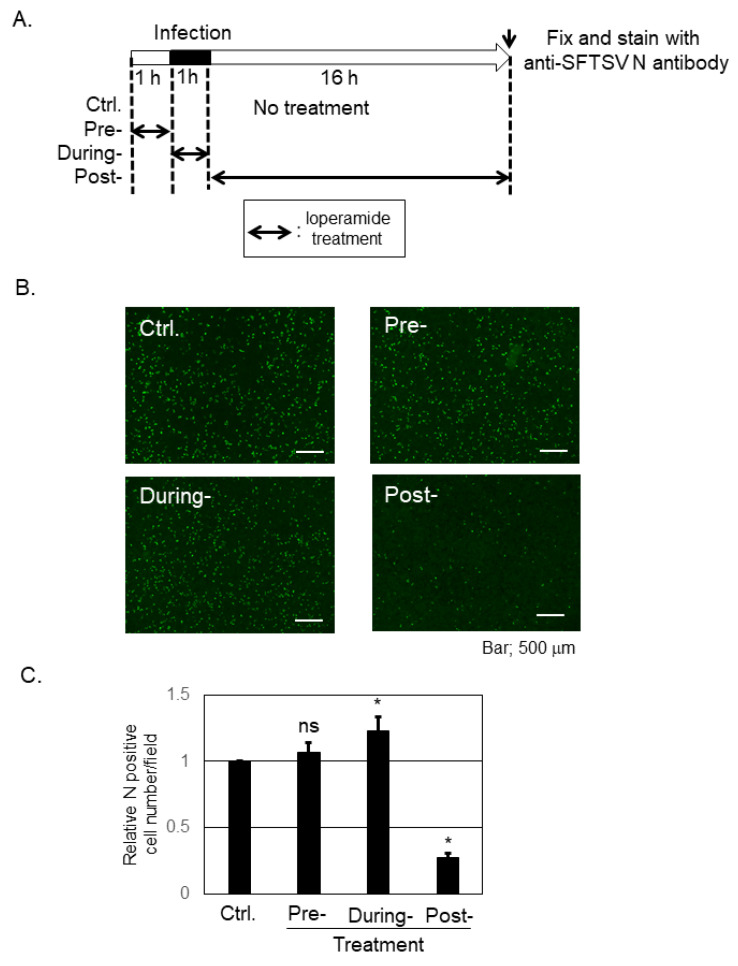
Loperamide affected the post-entry step, but not the pre- and during-entry steps, of the SFTSV infection. (**A**) Schematic representation of the time-of-addition infection assay, used to study the effect of loperamide on SFTSV infection. SFTSV was infected at an MOI of 1. (**B**,**C**) Loperamide-treated cells were fixed at 17 h p.i. (1 h for infection and 16 h for incubation) and stained with anti-SFTSV N antibody, followed by a secondary, FITC-conjugated anti-rabbit-IgG antibody. Images of stained cells were captured using a BZ-X700 microscope (**B**) and analyzed using the BZ-X analyzer software to automatically count the FITC-positive cell number in the fields. Four different fields were randomly selected from three independent wells to count the cells. Normalized FITC-positive cell numbers with control treatment were shown (**C**). Data correspond to the mean ± SD (ns; not significant, * *p* < 0.05).

**Figure 3 viruses-13-00869-f003:**
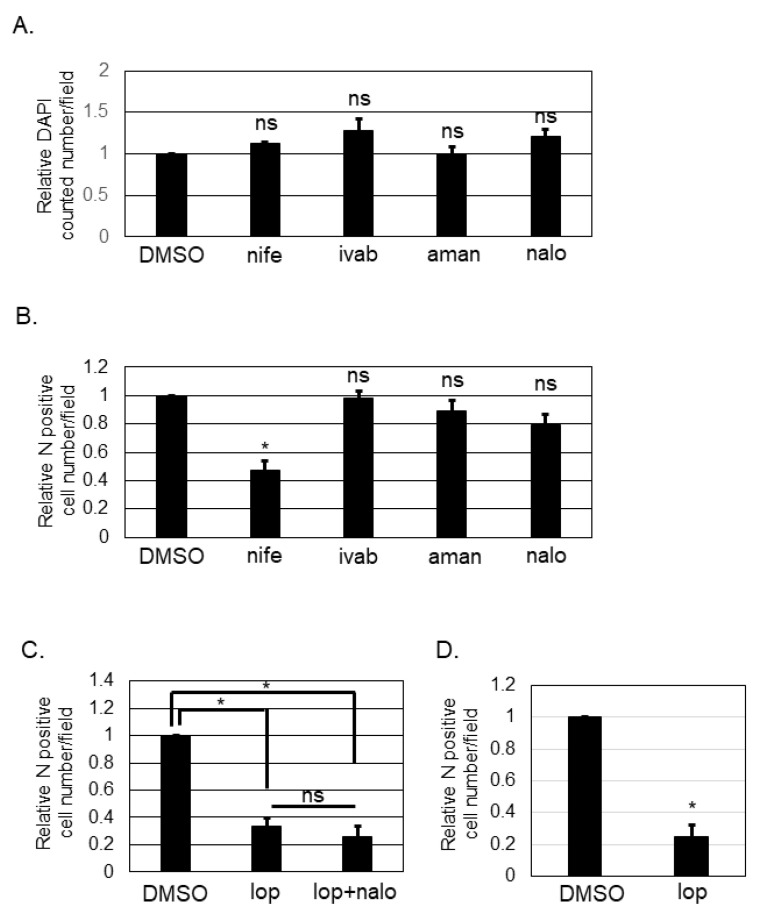
Identification of the mechanism of action of loperamide on the post-entry step of SFTSV infection. (**A**) Relative DAPI-positive cell numbers upon nifedipine (nife, 250 μM), ivabradine (ivab, 250 μM), amantadine (aman, 625 μM), or naloxone (nalo, 1 mM) treatments compared to DMSO treatment, in Huh-7 cells at 24 h post-treatment. (**B**) Relative SFTSV N-positive cell numbers per field at 16 h p.i. in Huh-7 cells. (**C**) Relative SFTSV N-positive cell numbers per field upon DMSO, loperamide (lop, 20 μM), or lop (20 μM) with nalo (1 mM) treatment at 16 h post-treatment. (**D**) Vero 76 cells were infected with SFTSV and treated with either DMSO or loperamide (20 μM). Relative N-positive cell numbers per field at 24 h p.i. were shown. Data correspond to the mean ± SD (ns; not significant, * *p* < 0.05).

**Figure 4 viruses-13-00869-f004:**
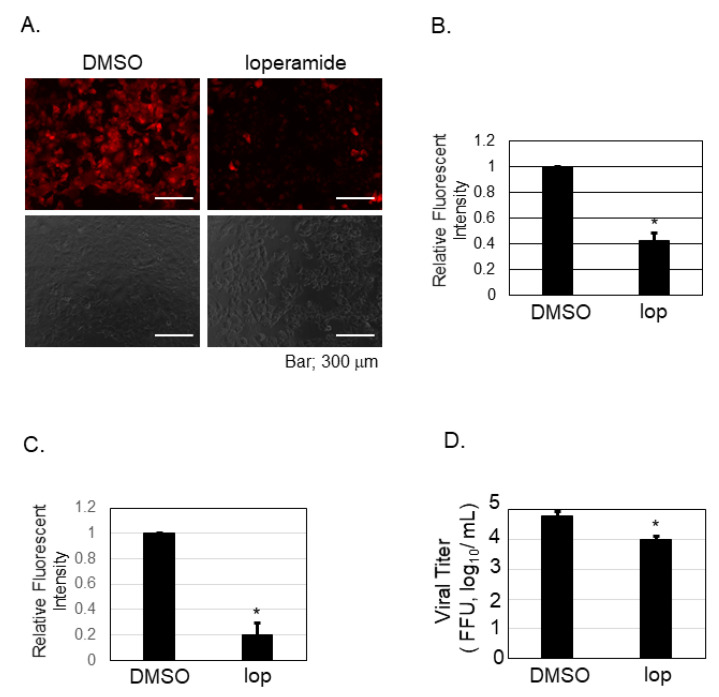
Calcium imaging and analysis upon loperamide treatment in Huh-7 cells. Huh-7 cells were treated with either DMSO or 10 μM of loperamide (lop) and incubated for 48 h in the absence (**A**,**B**) or in the presence (**C**,**D**) of SFTSV. (**A**) Culture supernatant of Huh-7 cells was replaced with Rhod-4AM (10 μM) diluted in Opti-MEM and incubated for further 20 min at 37 °C in 5% CO_2_. Fluorescent and bright field images of the fixed cells were captured. (**B**) Fluorescence intensity was measured and the relative mean ± SD (***
*p* < 0.05) is shown. (**C**) Same treatment described in (**A**), but infected with SFTSV (moi = 0.1), was performed. Fluorescence intensity was measured and the relative mean ± SD (***
*p* < 0.05) is shown. (**D**) The supernatant of the cells measured in (**C**) was collected to measure the viral titration. Data correspond to the mean ± SD (* *p* < 0.05).

## Data Availability

Not applicable.

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
