# Peer review of "Loperamide Inhibits Replication of Severe Fever with Thrombocytopenia Syndrome Virus"

_viruses, 2021, doi:10.3390/v13050869_

Round 1

Reviewer 1 Report

Re:Manuscript ID: viruses-1166785
Title: Loperamide Targets Replication Step of Severe Fever with
Thrombocytopenia Syndrome Virus

Urata S. et al describe an inhibitory effect of Loperamide on SFTS virus replication by in vitro virus culture studies. The proposed mechanism underlying SFTSV inhibition by this FDA-approved drug is a reduced calcium influx into the infected cells. The findings are interesting and the data are clearly presented. However, there are several minor points that need to be clarified before publication.

Minor points:

1.Title: the experiments seem not to be focused on virus “replication step”. The title can be simply changed to “Loperamide inhibits replication of Severe Fever with Thrombocytopenia Syndrome Virus”

  1. Line 65, Please describe why Huh7 and SW13 were selected for the inhibition assay.
  2. Line 119, according to the figure 1AB, I think the reduction rate in Huh-7 at 48h is approximately 20. Also, the reduction rate in SW13 at 48h is approximately 100.
  3. Figure 1C and 1D, in order to see cytotoxic effects of lop, did the authors perform the experiment of lop treatment only (w/o SFTSV infection)? Is the caption for this figure correct? Also, did the authors observe any cytotoxic effect on Huh7 at 48h?
  4. Line 233, the “twenty-five times lower” is unclear.

Author Response

Reviewer#1

Urata S. et al describe an inhibitory effect of Loperamide on SFTS virus replication by in vitro virus culture studies. The proposed mechanism underlying SFTSV inhibition by this FDA-approved drug is a reduced calcium influx into the infected cells. The findings are interesting and the data are clearly presented. However, there are several minor points that need to be clarified before publication.

Thank you for your time to review our manuscript and fruitful suggestion to improve our manuscript. We sincerely considered the points raised by the reviewer and responded as below.

Q1. Title: the experiments seem not to be focused on virus “replication step”. The title can be simply changed to “Loperamide inhibits replication of Severe Fever with Thrombocytopenia Syndrome Virus”

A1. As reviewer suggested, we changed our title as “Loperamide inhibits replication of Severe Fever with Thrombocytopenia Syndrome Virus” as in the title of the manuscript (page 1, lines 2-3).

Q2. Line 65, Please describe why Huh7 and SW13 were selected for the inhibition assay.

A2. Thank you for your comment. It was reported that SFTSV infected cells could be detected in several organs from patients, including liver and adrenal gland (Suzuki et al., 2020). We selected two human organ originated cell lines, Huh-7 from hepatocyte and SW13 from adrenal cortex, to examine if the anti-SFTSV effect of Loperamide is cell type specific or the general outcome. This explanation was added in page 5, lines 125-126.

Q3. Line 119, according to the figure 1AB, I think the reduction rate in Huh-7 at 48h is approximately 20. Also, the reduction rate in SW13 at 48h is approximately 100.

A3. As reviewer pointed out, we corrected our original manuscript (page 5, lines 134 and 136).

Q4. Figure 1C and 1D, in order to see cytotoxic effects of lop, did the authors perform the experiment of lop treatment only (w/o SFTSV infection)? Is the caption for this figure correct? Also, did the authors observe any cytotoxic effect on Huh7 at 48h?

A4. Thank you for the comment. As described in the main text, the results of the relative DAPI number shown in Figure 1C and 1D were from the infected cells. To avoid confusion, the sentence was re-written (page 5, lines 138-139). In terms of the cytotoxicity of the loperamide on Huh-7 cells at 48 h post treatment, slightly but statistically significant reduction (approximately 11.4 %) was observed compared to the control treatment as described in page 6 line 140 and Figure 1D. However, we concluded that loperamide inhibited the SFTSV production based on the results that loperamide treatment reduced SFTSV production (twenty five-folds reduction) significantly, compared to the control treatment (page 13, lines 268-271).

Q

Reviewer#1

Urata S. et al describe an inhibitory effect of Loperamide on SFTS virus replication by in vitro virus culture studies. The proposed mechanism underlying SFTSV inhibition by this FDA-approved drug is a reduced calcium influx into the infected cells. The findings are interesting and the data are clearly presented. However, there are several minor points that need to be clarified before publication.

Thank you for your time to review our manuscript and fruitful suggestion to improve our manuscript. We sincerely considered the points raised by the reviewer and responded as below.

Q1. Title: the experiments seem not to be focused on virus “replication step”. The title can be simply changed to “Loperamide inhibits replication of Severe Fever with Thrombocytopenia Syndrome Virus”

A1. As reviewer suggested, we changed our title as “Loperamide inhibits replication of Severe Fever with Thrombocytopenia Syndrome Virus” as in the title of the manuscript (page 1, lines 2-3).

Q2. Line 65, Please describe why Huh7 and SW13 were selected for the inhibition assay.

A2. Thank you for your comment. It was reported that SFTSV infected cells could be detected in several organs from patients, including liver and adrenal gland (Suzuki et al., 2020). We selected two human organ originated cell lines, Huh-7 from hepatocyte and SW13 from adrenal cortex, to examine if the anti-SFTSV effect of Loperamide is cell type specific or the general outcome. This explanation was added in page 5, lines 125-126.

Q3. Line 119, according to the figure 1AB, I think the reduction rate in Huh-7 at 48h is approximately 20. Also, the reduction rate in SW13 at 48h is approximately 100.

A3. As reviewer pointed out, we corrected our original manuscript (page 5, lines 134 and 136).

Q4. Figure 1C and 1D, in order to see cytotoxic effects of lop, did the authors perform the experiment of lop treatment only (w/o SFTSV infection)? Is the caption for this figure correct? Also, did the authors observe any cytotoxic effect on Huh7 at 48h?

A4. Thank you for the comment. As described in the main text, the results of the relative DAPI number shown in Figure 1C and 1D were from the infected cells. To avoid confusion, the sentence was re-written (page 5, lines 138-139). In terms of the cytotoxicity of the loperamide on Huh-7 cells at 48 h post treatment, slightly but statistically significant reduction (approximately 11.4 %) was observed compared to the control treatment as described in page 6 line 140 and Figure 1D. However, we concluded that loperamide inhibited the SFTSV production based on the results that loperamide treatment reduced SFTSV production (twenty five-folds reduction) significantly, compared to the control treatment (page 13, lines 268-271).

Q5. Line 233, the “twenty-five times lower” is unclear.

A5. As reviewer pointed out, we rephrased the sentence (page 13, line 269).

5. Line 233, the “twenty-five times lower” is unclear.

A5. As reviewer pointed out, we rephrased the sentence (page 13, line 269).

Reviewer 2 Report

The manuscript of Urata et al. describes the effect of loperamide on the replication of Severe Fever with Thrombocytopenia Syndrome Virus (SFTSV). Even though the work described here has potentially interesting therapeutic applications, in its present state it lacks sufficiently solid data to reach any conclusion. I therefore do not recommend publication.

Figure 1:

-What is the relevance of the cell lines used here with respect to SFTSV-induced cytopathology ?

-Viral titration is not a direct measure of viral replication but a measure of infecting particles. Viral replication needs to be analyzed by determining the expression of viral RNA throughout the time of the experiment.

At an MOI=0.1, the production of new viral particles depends on the capacity of the virus not only to replicate but also to propagate from the primo infected cells to neighboring cells. Viral propagation depends among others on the capacity of the host cell to set up an efficient antiviral response. The kinetics of induction of the type interferon response induced by SFTSV in the presence or absence of loperamide needs to be analyzed.

-DAPI staining is by no means a test of cytotoxicity: conditions that affect cell division will affect DAPI staining in the absence of cytotoxicity. Authors need to use vital dyes in order to measure cytotoxicity.

Figure 2:

-Why an MOI=1 is used here whereas an MOI=0.1 was used in figure 1 ?

-Pre- and during-treatment led to an increase in the number of N positive cells. This seems contradictory with previous statements indicative that loperamide treatment leads to an inhibition of viral replication. Authors need to clarify this point.

-To what corresponds the post-entry step ? Is this different than viral replication ?

-Why the number of N positive cells, that depends not only on viral replication but also the % of infectivity, was measured here and not viral titers as in Fig 1 ?

-Viral titers and viral replication need to be measured.

Figure 3:

-Viral titers and viral replication need to be measured.

Figure 4:

-The effect of loperamide on calcium influx has to be measured in the presence and absence of SFTSV.

Finally, in vivo data should certainly strengthen the work. 

Author Response

Reviewer#2

The manuscript of Urata et al. describes the effect of loperamide on the replication of Severe Fever with Thrombocytopenia Syndrome Virus (SFTSV). Even though the work described here has potentially interesting therapeutic applications, in its present state it lacks sufficiently solid data to reach any conclusion. I therefore do not recommend publication.

Thank you for your time and the comments. We believe that our conclusion that the loperamide inhibited SFTSV propagation, especially at the post-entry step of infection, and this inhibition was due to the prevention of the calcium influx was supported by our results. We performed some experiments suggested by the reviewer to strengthen our conclusion as described below.

Figure 1:

Q1. What is the relevance of the cell lines used here with respect to SFTSV-induced cytopathology ?

A1. Thank you for your comment. We selected two human organ originated cell lines, Huh-7 from hepatocyte and SW13 from adrenal cortex, to examine if the anti-SFTSV effect of loperamide is cell type specific or the general outcome. We did not observe any cytopathic effect due to the infection with both cell lines, which is consistent with our previous report (Urata et al., 2018).

Q2. Viral titration is not a direct measure of viral replication but a measure of infecting particles. Viral replication needs to be analyzed by determining the expression of viral RNA throughout the time of the experiment.

A2. We concur with the reviewer that viral replication was not measured in our experiment. Therefore, the title was changed to “Loperamide inhibits replication of Severe Fever with Thrombocytopenia Syndrome Virus”. In addition, we changed the phrase from viral replication to the post-entry step of SFTSV infection throughout the text.

Q3. At an MOI=0.1, the production of new viral particles depends on the capacity of the virus not only to replicate but also to propagate from the primo infected cells to neighboring cells. Viral propagation depends among others on the capacity of the host cell to set up an efficient antiviral response. The kinetics of induction of the type interferon response induced by SFTSV in the presence or absence of loperamide needs to be analyzed.

 A3. We concur with the reviewer. To assess reviewer’s point, Vero cells, which are known to be deficient for the type I interferon production, was infected with SFTSV, and treated in the presence or in the absence of the loperamide. As shown in Figure 3D, loperamide treatment on Vero cells reduced the SFTSV N positive cells compared to the DMSO treatment, suggesting that the loperamide could inhibit the post-entry step of SFTSV infection without the type I interferon response.

Q4. DAPI staining is by no means a test of cytotoxicity: conditions that affect cell division will affect DAPI staining in the absence of cytotoxicity. Authors need to use vital dyes in order to measure cytotoxicity.

 A4. As reviewer pointed out, DAPI staining does not always reflect the cytotoxicity. In our experiments, cells were seeded with confluent monolayers to minimize the effect of the cell division (page 3, line 74).

Figure 2:

Q5. Why an MOI=1 is used here whereas an MOI=0.1 was used in figure 1 ?

A5. In Figure 1, the effect of the loperamide on the SFTSV propagation was examined (multiple rounds of infection). In contrast, the effect from the pre-entry, during-entry, and the post-entry step was examined in Figure 2 (single round of infection). To obtain sufficient SFTSV N positive cells in the single round of infection, MOI = 1 was used for Figure 2.

Q6. Pre- and during-treatment led to an increase in the number of N positive cells. This seems contradictory with previous statements indicative that loperamide treatment leads to an inhibition of viral replication. Authors need to clarify this point.

A6. Thank you for the comment. As reviewer pointed out, we observed slight but significant increase of the N positive cell number when the loperamide was present 1 hour during virus adsorption. It is possible that loperamide non-specifically facilitates the entry-step of the SFTSV infection by, for instance, changing virion surface charge favorable for cell entry. However, we did not further assess this point, because the magnitude of the increase was small (22% increase) when compared with 3.6 folds reduction in N positive cell number with the loperamide post-treatment as mentioned in page 13, lines 276-281. This observation was consistent with the significant reduction of viral production in the presence of loperamide after virus adsorption until the end of the experiment (Figure 1A and 1B).

Q7. what corresponds the post-entry step ? Is this different than viral replication ?

A7. Post-entry step in our experiment includes viral RNA replication, transcription, and the translation of the viral protein. To better explain this term, we added an explanation at the page 7, line 169-170.

Q8. Why the number of N positive cells, that depends not only on viral replication but also the % of infectivity, was measured here and not viral titers as in Fig 1 ?

A8. The rationale of this experiment was to examine the target of the loperamide against the SFTSV infection. Viral production shown in Figure 1 was the result of the entry, genome replication, gene transcription, translation, assembly/budding, and viral production. To focus on the early- and middle-replication steps of the virus infection, we did not measure the viral titer, but counted N positive cells.

Q9. Viral titers and viral replication need to be measured.

A9. The rationale of this experiment was to examine if loperamide affected to the pre-entry (attachment), during-entry (internalization), or post-entry (replication to protein translation) steps. In fact, the viral production upon loperamide post-infection treatment was shown in Figure 1A for Huh-7 cells.

Figure 3:

Q10. Viral titers and viral replication need to be measured.

 A10. The major aim of this experiment was to show the compounds’ effect at the post-entry step (viral replication to the protein translation), which was the main target of the loperamide as shown in Figure 2. Therefore, we believe that the result of the viral titer and the viral replication is not absolutely required.

Figure 4:

Q11. The effect of loperamide on calcium influx has to be measured in the presence and absence of SFTSV.

A11. In addition to the result showing the effect of loperamide on calcium influx in the absence of SFTSV, the effect of loperamide on calcium influx was measured in the presence of the SFTSV (Figure 4C in the new manuscript). The culture supernatant of the samples, which were used to measure calcium influx, was collected to measure the SFTSV titer (Figure 4D). As a result, it was shown that loperamide inhibited calcium influx both in the presence and in the absence of the SFTSV.

Q12. Finally, in vivo data should certainly strengthen the work. 

A12. We concur that in vivo data would strengthen our work. However, the main purpose of this study was to examine the effect of loperamide and its antiviral mechanism in vitro. We would surely examine the effect of loperamide in vivo i

Round 2

Reviewer 2 Report

The manuscript of Urata et al. describes the effect of loperamide on the replication of Severe Fever with Thrombocytopenia Syndrome Virus (SFTSV). Several of the main questions I raised after reading the first version of the manuscript have remained unanswered in the revised version. I am therefore not very enthusiastic about recommending publication but leave the final decision to the editor.

Comment 1

Since one of the main points of this work was to state that treatment with loperamide inhibited viral replication, I requested that besides measuring viral titers the authors measure the expression of viral RNA.

Notwithstanding my request, this point has remained unaddressed in the revised version while the answer of the authors to this question is quite puzzling:

We concur with reviewer that viral replication was not measured in our experiment. Therefore the title has been changed to “Loperamide inhibits replication of Severe Fever with Thrombocytopenia Syndrome Virus”.

The previous title was “Loperamide targets replication step…”. For me, the statement “inhibits replication” emphasizes even more strongly an effect of loperamide on viral replication and this even though viral replication was not measured as admitted by the authors and despite my request.

Comment 2

I pointed out that at MOI=0.1 production of new infectious particles (viral titers) depends among others on viral propagation from primo infected cells to non-infected neighboring cells, which itself depends on the capacity of the infected cell to induce an efficient type I interferon antiviral response. I therefore requested that the authors measure the induction of the type I IFN response before and after treatment with loperamide.

Despite my request, the measurement of the type I IFN response has remained unaddressed.

Instead authors provide staining of protein N in Vero cells (Fig.3D)… but staining of protein N is not comparable to measuring viral titers (Fig.1) and still does not answer to the question of the effect of loperamide on type I IFN response.

Another way to circumvent the role of propagation on measurements of viral titers at MOI=0.1 could have been to measure viral titers at MOI>1 but this was not done either.

Author Response

Comment 1

Since one of the main points of this work was to state that treatment with loperamide inhibited viral replication, I requested that besides measuring viral titers the authors measure the expression of viral RNA.

Notwithstanding my request, this point has remained unaddressed in the revised version while the answer of the authors to this question is quite puzzling:

We concur with reviewer that viral replication was not measured in our experiment. Therefore the title has been changed to “Loperamide inhibits replication of Severe Fever with Thrombocytopenia Syndrome Virus”.

The previous title was “Loperamide targets replication step…”. For me, the statement “inhibits replication” emphasizes even more strongly an effect of loperamide on viral replication and this even though viral replication was not measured as admitted by the authors and despite my request.

Response 1

Thank you for your comment. As reviewer commented, we responded to the reviewer’s comment with changing out title. Since we did not perform the quantitative analysis of the replication steps, we removed “step” from our title.

Comment 2

I pointed out that at MOI=0.1 production of new infectious particles (viral titers) depends among others on viral propagation from primo infected cells to non-infected neighboring cells, which itself depends on the capacity of the infected cell to induce an efficient type I interferon antiviral response. I therefore requested that the authors measure the induction of the type I IFN response before and after treatment with loperamide.

Despite my request, the measurement of the type I IFN response has remained unaddressed.

Instead authors provide staining of protein N in Vero cells (Fig.3D)… but staining of protein N is not comparable to measuring viral titers (Fig.1) and still does not answer to the question of the effect of loperamide on type I IFN response.

Another way to circumvent the role of propagation on measurements of viral titers at MOI=0.1 could have been to measure viral titers at MOI>1 but this was not done either.

Response 2

We concur with the reviewer’s concern. Therefore, to examine if the type I IFN is involved in the antiviral effect of loperamide or not, we used Vero cell lines, which are known to be defective for the Type I IFN production (Figure 3D). Our result clearly showed that the reduction of the SFTSV replication was observed without a type I IFN production.